# Selection signatures in goats reveal copy number variants underlying breed-defining coat color phenotypes

Jan Henkel[1,2], Rashid Saif[1,3], Vidhya Jagannathan[1,2], Corinne Schmocker[1], Flurina Zeindler[4], Erika Bangerter[5], Ursula Herren[5], Dimitris Posantzis[6], Zafer Bulut[7], Philippe Ammann[8], Cord Drögemüller[1,2], Christine Flury[4], Tosso Leeb[1,2]*

1 Institute of Genetics, Vetsuisse Faculty, University of Bern, Bern, Switzerland, 2 DermFocus, University of Bern, Bern, Switzerland, 3 Institute of Biotechnology, Gulab Devi Educational Complex, Lahore, Pakistan, 4 School of Agricultural, Forest and Food Sciences, Bern University of Applied Sciences, Zollikofen, Switzerland, 5 Swiss Goat Breeding Association, Zollikofen, Switzerland, 6 Attica Zoological Park, Spata, Greece, 7 Department of Biochemistry, Faculty of Veterinary Medicine, Selcuk University, Konya, Turkey, 8 ProSpecieRara, Basel, Switzerland

* Tosso.Leeb@vetsuisse.unibe.ch

**Data Availability Statement:** All relevant data are within the manuscript and its Supporting Information files.

## Abstract

Domestication and human selection have formed diverse goat breeds with characteristic phenotypes. This process correlated with the fixation of causative genetic variants controlling breed-specific traits within regions of reduced genetic diversity, so called selection signatures or selective sweeps. Using whole genome sequencing of DNA pools (pool-seq) from 20 genetically diverse modern goat breeds and bezoars, we identified 2,239 putative selection signatures. In two Pakistani goat breeds, Pak Angora and Barbari, we found selection signatures in a region harboring *KIT*, a gene involved in melanoblast development, migration, and survival. The search for candidate causative variants responsible for these selective sweeps revealed two different copy number variants (CNVs) downstream of *KIT* that were exclusively present in white Pak Angora and white-spotted Barbari goats. Several Swiss goat breeds selected for specific coat colors showed selection signatures at the *ASIP* locus encoding the agouti signaling protein. Analysis of these selective sweeps revealed four different CNVs associated with the white or tan ($A^{Wt}$), Swiss markings ($A^{sm}$), badger-face ($A^b$), and the newly proposed peacock ($A^{pc}$) allele. RNA-seq analyses on skin samples from goats with the different CNV alleles suggest that the identified structural variants lead to an altered expression of *ASIP* between eumelanistic and pheomelanistic body areas. Our study yields novel insights into the genetic control of pigmentation by identifying six functionally relevant CNVs. It illustrates how structural changes of the genome have contributed to phenotypic evolution in domestic goats.

## Author summary

Domestic animals have been selected for hundreds or sometimes even thousands of years for traits that were appreciated by their human owners. This process correlated with the

**Funding:** This study was funded by a grant from the Swiss National Science Foundation (31003A_172964). R.S. was supported by a Swiss Government Excellence Scholarship and a supplementary grant from the Hans Sigrist Foundation. The funders had no role in study design, data collection and analysis, decision to publish, or preparation of the manuscript.

**Competing interests:** The authors have declared that no competing interests exist.

fixation of causative genetic variants controlling breed-specific traits within regions of reduced genetic diversity, so called selection signatures or selective sweeps. We conducted a comprehensive screen for selection signatures in 20 phenotypically and genetically diverse modern goat breeds and identified a total of 2,239 putative selection signatures in our dataset. Follow-up experiments on selection signatures harboring known candidate genes for coat color revealed six different copy number variants (CNVs). Two of these CNVs were located in the 3'-flanking region of *KIT* and associated with a completely white coat color phenotype in Pak Angora goats and a white-spotted coat color phenotype in Barbari goats, respectively. The other four CNVs were located at the *ASIP* locus. They were associated with four different types of coat color patterning in seven Swiss goat breeds. Their functional effect is mediated by region-specific quantitative changes in *ASIP* mRNA expression. Our study illustrates how structural changes of the genome have contributed to phenotypic evolution in domestic goats.

## Introduction

Goat domestication started around 10,000 years ago in the fertile crescent and is believed to be one of the earliest domestication events of livestock animals [1, 2]. Bezoars, the wild ancestors of domestic goats are an extant species with a distribution in Western Asia from Turkey to Pakistan. Since domestication, goats followed the human migration [3] and played an economically important role for their owners by providing various products like milk, meat or fibers. These economical values were further increased by production-orientated breeding, which led to more than 600 diverse goat breeds at present time [4–6].

Artificial selection of domesticated goats not only resulted in specialized elite breeds for milk, meat or fibers, but also in breeds with unique coat color phenotypes [4, 7]. Due to their striking appearance, these goat breeds are of special value to their owners, selected for uniform coat color, and kept in closed populations. Coat color phenotypes are one of the most intensively studied traits in goats [8–12]. They include solid colored animals of different color, animals with symmetrical color patterns, and animals with white markings, white spotting phenotypes or completely white animals.

White markings, white spotting and completely white phenotypes typically result from a lack of melanocytes in the skin and hair follicles. This group of phenotypes is also termed leucism or piebaldism and characterized by defects in melanoblast development or migration [13–17].

Very light coat colors resembling white are also seen in animals that have a normal set of melanocytes synthesizing a very pale pheomelanin [18]. Melanocytes produce two types of pigments, the brown to black eumelanin and the red to yellow pheomelanin. The so-called pigment type switching, an intensively studied signaling process, governs whether a given melanocyte produces eumelanin or pheomelanin [19]. Eumelanin is produced, if MC1R is activated by its ligand α-melanocyte stimulating hormone (α-MSH), while pheomelanin is produced if α-MSH is absent and/or outcompeted by binding of the competitive antagonist ASIP to MC1R [20–25]. Different alternative promoters of the *ASIP* gene enable spatially and temporally regulated *ASIP* expression, which results in characteristic patterns of eumelanin and pheomelanin synthesis [25–28].

Domestication and artificial selection correlated with the fixation of causative genetic variants controlling breed-specific traits within regions of reduced genetic diversity, so called selection signatures or selective sweeps [29–31]. A method detecting regions of low

heterozygosity from sequence data of pooled individuals (pool-seq) was developed and used to identify loci under selection in chicken, pigs, rabbits and the Atlantic herring [32–35]. Recently, pooled heterozygosity scores were also applied to monitor loci under selection in goats [36].

In the present study, we aimed to gain a better understanding of the genetic variants determining breed-specific coat color phenotypes in goats. We therefore performed a comprehensive screen for selection signatures in bezoars and 20 breeds of domesticated goats.

## Results

### Selection signature analysis

For the present study, we selected 8–12 animals each from 20 phenotypically diverse domesticated goat breeds and their wild ancestor, the bezoar. We isolated genomic DNA from these animals and prepared equimolar DNA pools for sequencing (Table 1).

We obtained 2x150 bp paired-end sequence data corresponding to 30x genome coverage per pool, called high confidence SNVs and calculated pooled heterozygosity scores ($-ZH_P$) in 150 kb sliding windows with 75 kb step size (S1 Fig; S1 and S2 Tables). The significance threshold was conservatively set at $-ZH_P \geq 4$, which identified 5,220 windows with extremely reduced heterozygosity (0.8% of all windows). Overlapping windows were further merged into 2,239 selection signatures (1.1% of total genomic length). This corresponded to 112 selection signatures per breed pool on average (median = 81; S3 Table).

To evaluate the validity of the pool-seq approach, we compared the results from pool-seq data to individual whole genome sequence data of 120 goats from five different Swiss breeds (S1 Table). We called SNVs from the individual sequence data and calculated $H_P$ and $-ZH_P$

**Table 1. Breeds collected for pool sequencing (pool-seq).**

| Breed | Abbreviation | Breed origin | Animals per pool |
|---|---|---|---|
| Pak Angora | ANG | Pakistan | 10 |
| Appenzell goat | APZ | Switzerland | 12 |
| Barbari | BAR | Pakistan | 12 |
| Beetal | BEE | Pakistan | 12 |
| Bezoar (*Capra aegagrus*) | BEZ | wild ancestor | 8 |
| Grisons Striped goat | BST | Switzerland | 12 |
| Boer goat | BUR | Africa | 12 |
| Capra Grigia | CAG | Switzerland | 12 |
| Dera Din Panah | DDP | Pakistan | 12 |
| Chamois Colored goat | GFG | Switzerland | 12 |
| Kamori | KAM | Pakistan | 12 |
| Nachi | NAC | Pakistan | 12 |
| Nera Verzasca | NER | Switzerland | 12 |
| Pahari | PAH | Pakistan | 12 |
| Peacock goat | PFA | Switzerland | 12 |
| Saanen goat | SAN | Switzerland | 12 |
| St. Gallen Booted goat | STG | Switzerland | 10 |
| Teddy | TED | Pakistan | 12 |
| Toggenburg goat | TOG | Switzerland | 12 |
| Valais Blackneck goat | VAG | Switzerland | 12 |
| African Dwarf goat | ZWZ | Africa | 12 |

scores respectively (S2 Table). The pool-seq dataset and the dataset with individual sequences yielded similar results (S2 Fig).

As a validation of the significance threshold, we inspected our data for selection signatures near known causative variants for breed-defining coat color traits. The characteristic brown coat color of the Toggenburg goat is caused by a missense SNV in the *TYRP1* gene, p.Gly496Asp [10]. Toggenburg goats showed the expected selection signature harboring the *TYRP1* gene with a $-ZH_p$ value of 4.88 (Fig 1; S3 Table).

In addition to the search for reduced heterozygosity, we calculated $F_{ST}$ values for each breed pool in a pairwise comparison with bezoars. The $F_{ST}$ analysis identified 847 selection signatures or 0.4% of the total genomic length (S1 and S3 Figs, S4 Table).

## CNVs at the *KIT* locus in two Pakistani goat breeds

The completely white Pak Angora and the white spotted Barbari breeds showed strong selection signatures harboring the *KIT* gene on chromosome 6 with $-ZH_p$ values of 7.20 and 4.56 (Fig 1; S3 Table). We searched for candidate causative variants within the signatures, but did not detect any coding variants in the *KIT* gene. However, visual inspection of the short read alignments revealed two different copy number variants (CNVs) downstream of the *KIT* gene in the Pak Angora and Barbari breeds (Fig 2).

Both CNVs started ~63 kb downstream of *KIT* and covered ~100 kb of the genome reference sequence without known coding DNA. The short read-alignments of read-pairs spanning the amplification breakpoints confirmed that the individual copies of the CNV were arranged in tandem in a head to tail orientation (S4 Fig). The Pak Angora allele consisted of a triplication of the 100 kb region. The Barbari allele represented a duplication of the same ~100 kb region with an additional 16,280 bp deletion in its central part. The read-pair information at the breakpoints indicated that the deleted part was replaced by a 22,702 bp genomic fragment from the 5'-flanking sequence of the *RASSF6* gene, which is located 19 Mb further downstream on the same chromosome (S4 Fig; S5 Table). The shared breakpoints of the two different CNV alleles suggested a common origin of these alleles.

## CNVs at the *ASIP* locus in Swiss goat breeds

Five Swiss goat breeds with different coat color patterns had a selection signature with $-ZHp \geq 4$ in the region of the *ASIP* locus on chromosome 13 (Fig 1; S3 Table). We did not find any *ASIP* coding variation in the breeds with *ASIP* selection signatures.

Based on the segregation of coat color patterns in a large breeding experiment, the existence of up to 11 different caprine *ASIP* alleles has been postulated [8]. The most dominant of this allelic series, termed "white or tan" ($A^{Wt}$), is responsible for white coat color in goats [8]. Furthermore, it has been shown that the white coat color in Saanen goats is caused by a triplication of the *ASIP* gene [9].

Inspection of the short-read alignments of our own sequence data from Saanen goats at the *ASIP* locus confirmed the previously reported triplication and revealed the exact boundaries of the triplication. It spans 154,677 bp of the reference genome sequence and comprises the entire coding sequence of the *ASIP*, *AHCY* and *ITCH* genes. The individual copies are arranged in tandem in a head to tail orientation. Appenzell goats, another white goat breed, had the same CNV allele as the Saanen goats (Fig 3, S4 Fig, S5 Table).

We next investigated the coverage plots of Grisons Striped goats and Toggenburg goats. These two breeds show a characteristic color pattern, which has been postulated to be caused by an *ASIP* allele termed "Swiss markings" ($A^{sm}$) [8]. They are fixed for an *ASIP* allele with 8

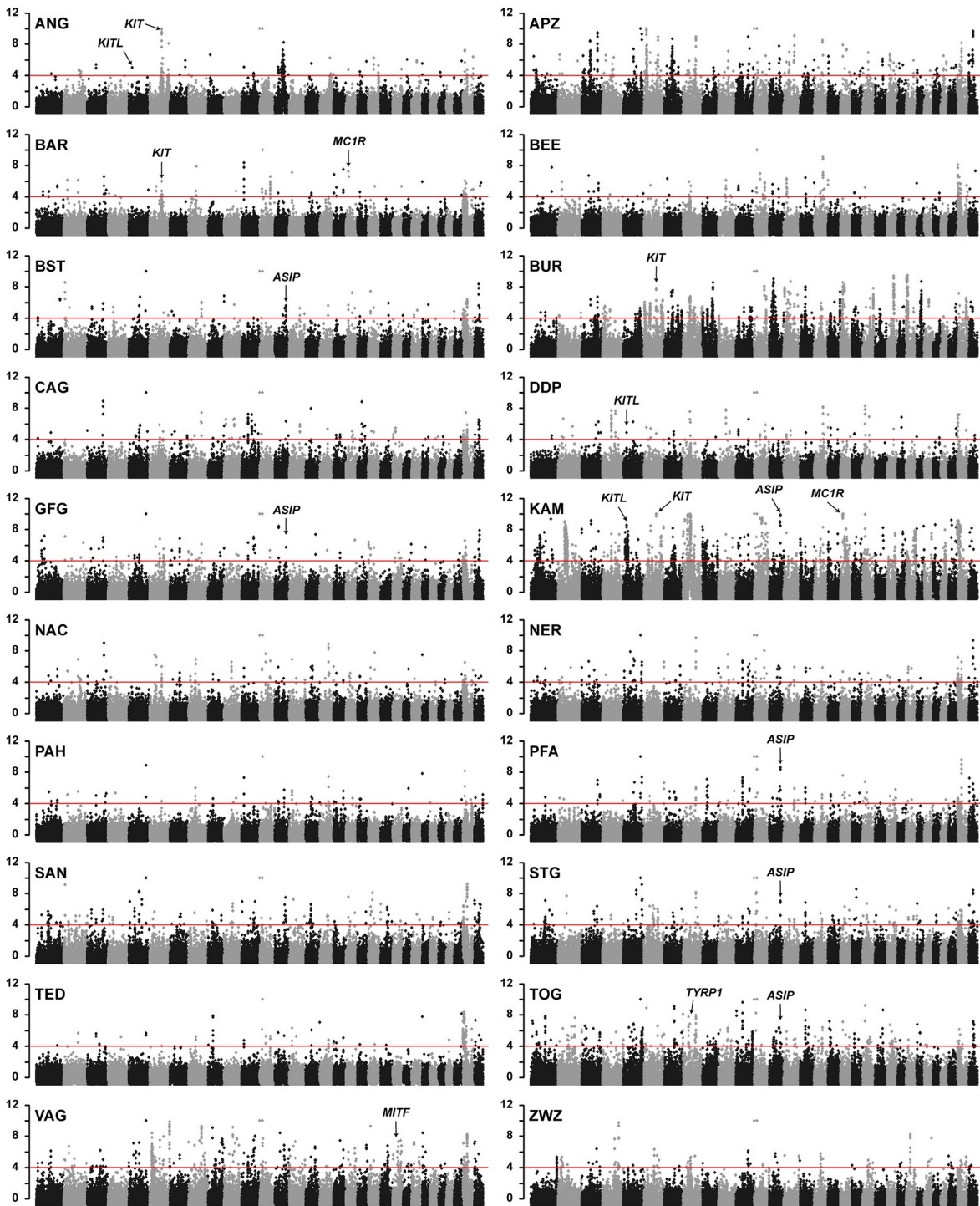

**Fig 1. Manhattan plots showing −ZHp values from 20 diverse goat breeds.** The red horizontal line indicates the chosen significance threshold of −ZHp = 4. Each dot represents a 150 kb window. Each plot contains 29 autosomes and two unplaced scaffolds representing the X chromosome. Selection signatures co-localizing with known coat color genes are marked with arrows.

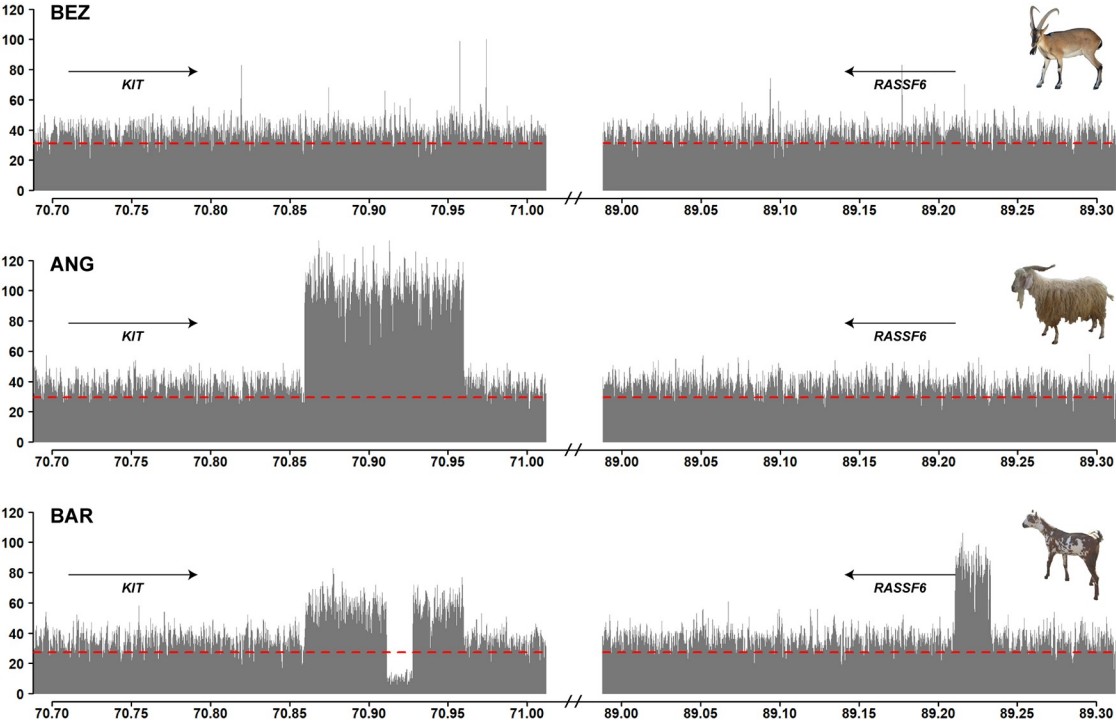

**Fig 2. CNVs at the *KIT* locus.** The coverage plot of bezoars (BEZ) does not show any copy number variation and represents the wildtype allele. In the Pak Angora breed (ANG), the coverage plot shows a triplication of ~100 kb downstream of the *KIT* gene. In the Barbari breed (BAR), the same region is duplicated. The Barbari allele shows a complex rearrangement involving the insertion of a ~23 kb genome segment originating at 89.2 Mb into the duplicated sequence at ~70.9 Mb with the simultaneous deletion of ~16 kb of *KIT* sequence. Please note that the coverage at ~89.2 Mb corresponds to three times the average. One genome equivalent corresponds to the wildtype sequence at ~89.2 Mb. Read-pair information indicated that the other two genome equivalents are inserted into the duplicated sequence at ~70.9 Mb (S4 Fig). The dashed red line indicates the average coverage across the whole genome of each pool-seq dataset.

tandem copies of a 13,433 bp sequence from the 5'-flanking region of *ASIP* (Fig 3, S4 Fig, S5 Table).

The Chamois Colored goat and the St. Gallen Booted goat are characterized by a color pattern and an *ASIP* allele termed "badgerface" ($A^b$) [8]. Our pool-seq data revealed a five-fold amplification of 45,680 bp located ~61 kb downstream of the $A^{sm}$ amplification (Fig 3, S4 Fig, S5 Table).

The Peacock goat is a rare Swiss goat breed with a unique and striking coat color pattern that has not been investigated previously. Pool-seq data from Peacock goats indicated a selection signature at the *ASIP* locus. The *ASIP* allele in Peacock goats, which we propose to term "peacock" ($A^{pc}$), has a quadruplication of the same ~45 kb region having five copies in the $A^b$ allele. It is additionally flanked by triplicated segments of 27,996 bp and 41,807 bp on the left and right side of the quadruplicated sequence (Fig 3, S4 Fig, S5 Table). The central part of the $A^{pc}$ allele has exactly the same breakpoints as the $A^b$ allele suggesting a common origin of $A^b$ and $A^{pc}$.

The goat genome reference sequence is derived from a San Clemente goat, which has a similar coat color pattern as bezoars. The genome reference therefore supposedly represents the wildtype allele at the *ASIP* locus, termed "bezoar" ($A^{bz}$) [8]. Bezoars and all other Swiss goat breeds did not show any CNVs at the *ASIP* locus. In the remaining goat breeds the $A^{Wt}$, $A^{sm}$ and $A^b$ alleles were segregating at low frequencies. These breeds are either not specifically

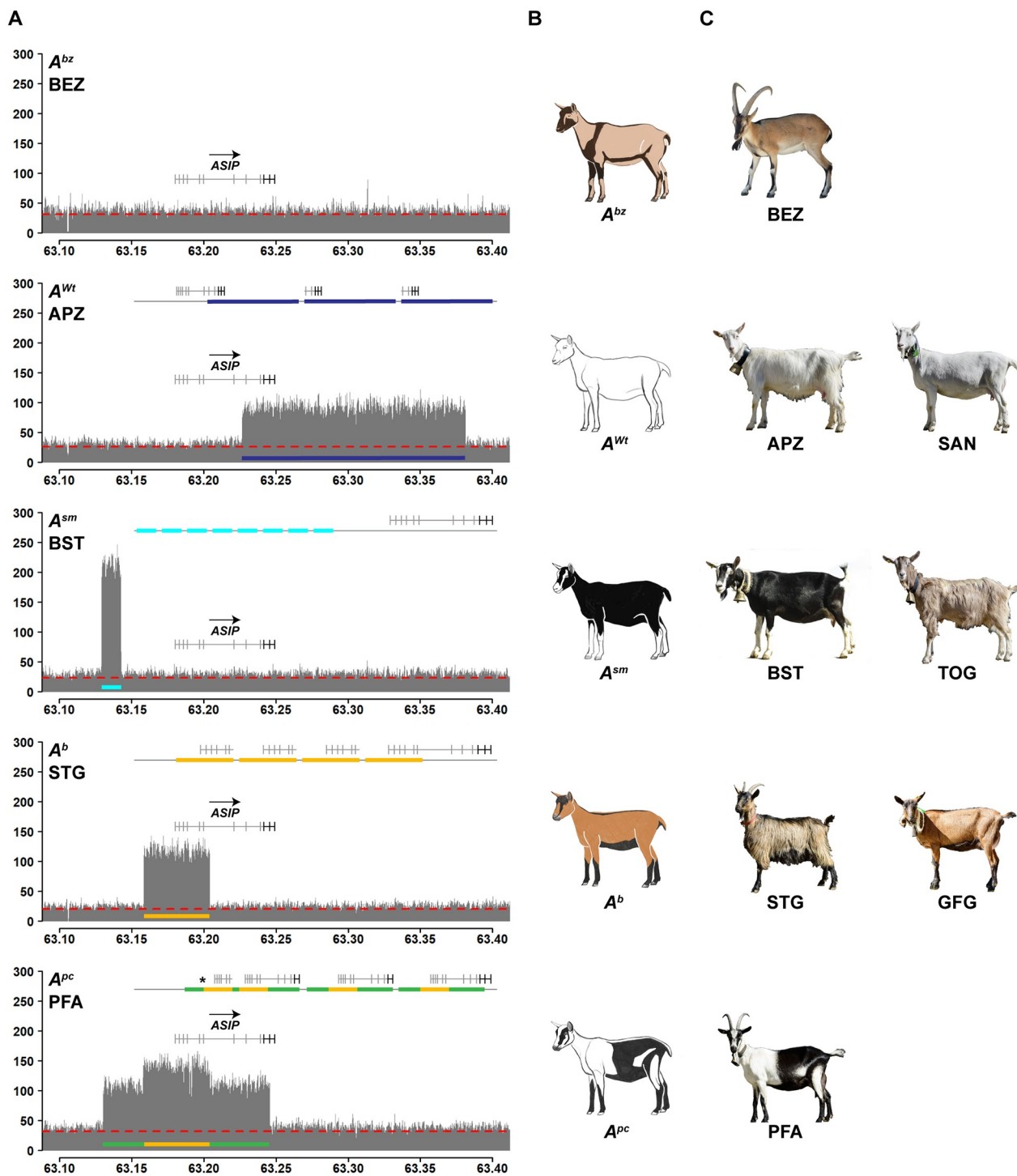

**Fig 3. CNVs at the *ASIP* locus. A** Coverage plots of the *ASIP* locus in different goat breeds reveal four different CNVs. The bezoar (BEZ) coverage plot shows uniform coverage and is characteristic for the wildtype allele ($A^{bz}$). Underneath, four different mutant *ASIP* alleles associated with different CNVs are illustrated. The line on top of each plot schematically indicates the most likely configuration of these mutant alleles derived from the available short-read sequence information (S4 Fig). The dashed red line indicates the average coverage across the whole genome of each breed. **B** Schematic drawings and **C** representative photographs illustrating the coat color phenotypes of the studied breeds. The photo of the bezoar was obtained during summer, when the dark stripes at the collar and the belly are much less pronounced than in the winter coat. Note that some of the patterns show an exactly inverse distribution of eumelanin and pheomelanin. For example, goats with the $A^{sm}$ allele have white (pheomelanistic) facial stripes and legs, while goats with the $A^b$ or $A^{pc}$ alleles have black (eumelanistic) facial stripes and legs.

selected for coat color (e.g. African dwarf goat, Beetal) or the effect of *ASIP* is phenotypically not visible due to epistatic effects of other genes that lead to white spotting phenotypes (e.g. Boer goat, Pak Angora, Barbari).

## Quantitative analysis of *ASIP* mRNA expression

The different CNVs in putative regulatory regions of the *ASIP* gene prompted us to hypothesize that quantitative differences in *ASIP* expression may cause the different color patterns. We therefore obtained whole skin samples from five goats carrying different CNV alleles. We isolated RNA from matched pairs of eumelanistic and pheomelanistic skin and performed an RNA-seq experiment to determine the expression level of *ASIP* mRNA expression.

In Grisons Striped goats ($A^{sm}$), Chamois Colored goats ($A^b$) and Peacock goats ($A^{pc}$), the eumelanistic skin showed very low *ASIP* mRNA expression. The pheomelanistic skin regions in these three goats had at least 10-fold higher *ASIP* expression than the corresponding eumelanistic samples. The uniformly white (pheomelanistic) Saanen goat ($A^{Wt}$) had the highest *ASIP* mRNA expression. There was no obvious correlation between the quantitative *ASIP* mRNA expression and the intensity of the pheomelanistic pigmentation. The intensely red colored skin from the Chamois Colored goat had an intermediate *ASIP* mRNA expression compared to the pale white skin from e.g. the Saanen and Peacock goat. Visual inspection of the RNA-seq short-read alignments indicated the utilization of nine different 5'-untranslated exons in nine different transcript isoforms originating from different *ASIP* alleles (Fig 4; S1 File).

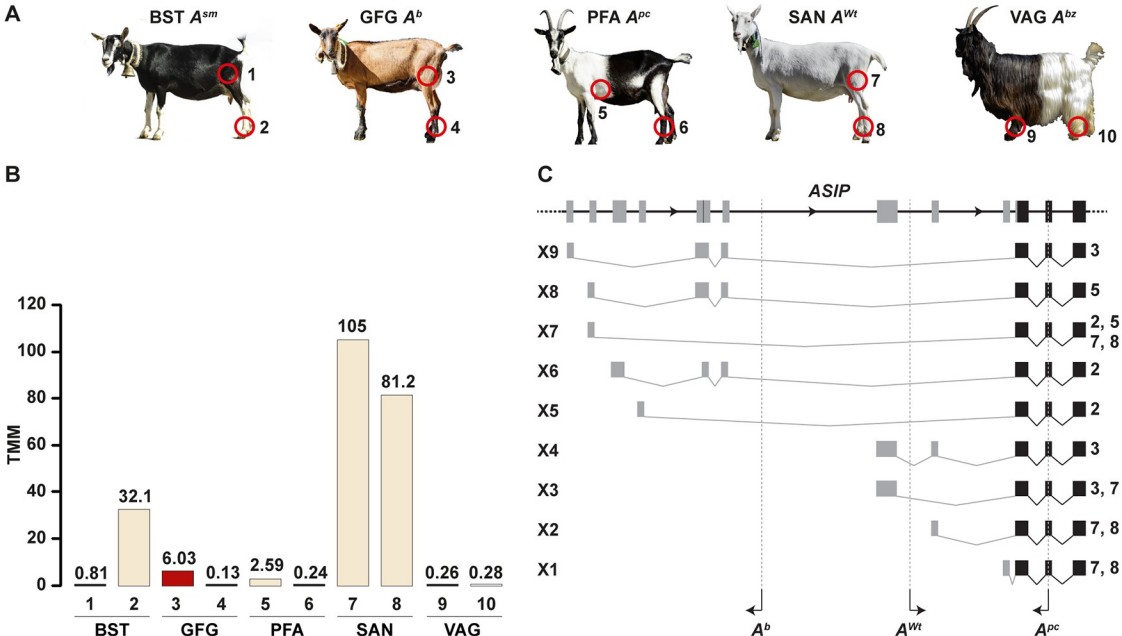

**Fig 4. *ASIP* mRNA expression and identified transcripts in skin. A** Representative photographs of the five sampled goat breeds. The biopsy sites are numbered and indicated by red circles. **B** Trimmed mean of M (TMM) values of *ASIP* mRNA expression were determined from RNA-seq data for each sample. The colors of the bars correspond to the pigmentation of the skin samples. Please note that the Valais Blackneck goat (VAG) has a black base color that is independent of the *ASIP* gene. This goat has a white spotting phenotype and lacks melanocytes in its caudal half. The low *ASIP* expression in the unpigmented white skin sample of this goat underscores the difference to the pheomelanistic pale white pigmentation in other goats. **C** *ASIP* transcript isoforms in pheomelanistic skin samples from goats with different *ASIP* alleles. Transcript isoforms X1 and X2 correspond to the RefSeq accessions XM_018057735.1 and XM_018057736.1. CNV breakpoints of the $A^b$, $A^{Wt}$, and $A^{pc}$ alleles are indicated.

## Discussion

In the present study, we discovered 2,239 loci under selection in 20 diverse goat breeds with various phenotypes and different geographical origins. Our methodology comprised the identification of regions with low heterozygosity from pool-seq data in combination with pairwise $F_{ST}$ to bezoars, the wild ancestor of domesticated goats. The pool-seq approach was validated by repeating the analyses with virtually identical results from individually sequenced goats in five breeds. We have to caution that reduced heterozygosity and high $F_{ST}$ values may not only result from selection, but may also be due to random demographic changes.

The comprehensive catalogue of identified selection signatures can now be used as a starting point to identify causal genetic variants that control a wide variety of breed-defining traits.

We particularly focused on selection signatures harboring known coat color genes and identified two CNVs in the 3'-flanking region of *KIT* in two Pakistani goat breeds, the completely white Pak Angora breed and the white spotted Barbari breed. An association between white coat color and the *KIT* locus has been reported before in Iranian Markhoz goats, which also represents an Angora type goat [37]. The *KIT* gene is flanked by several hundred kilobases of non-coding genomic DNA on either side, which are required for the precise regulation of its temporal and spatial expression. The KIT protein is a receptor tyrosine kinase mediating a survival signal for several different cell types including melanoblasts and melanocytes, but also e.g. hematopoetic stem cells, mast cells, interstitial cells of Cajal and spermatogonia [38,39]. *KIT* is a proto-oncogene and its overexpression may have detrimental consequences such as tumor development [40,41]. Insufficient expression of functional KIT protein in melanoblasts or melanocytes will lead to apoptosis of these cells and results in white spotting phenotypes [15,42–45].

Structural variants at the *KIT* locus cause several other breed defining coat color phenotypes in domestic animals, such as the dominant white and belt phenotypes in pigs [33,46,47], color-sided and lineback in cattle [48,49], and tobiano spotting in horses [50]. Lineback in cattle and tobiano in horses also involve structural variants in the 3'-flanking region of *KIT* [49,50]. All these phenotypes are characterized by striking alterations in pigmentation without any deleterious consequences on the other KIT dependent cell types, which would be expected to result in potentially serious health problems. The comparative data from other species strongly suggest that the newly detected caprine *KIT* CNVs in goats cause the complete lack of skin and hair pigmentation in Pak Angora and the white spotted phenotype in Barbari goats due to altered expression of KIT during fetal development of melanoblasts.

The *ASIP* gene codes for the agouti signaling protein, the competitive inhibitor of melanocortin 1 receptor expressed on melanocytes [20]. Variation in the quantitative amount of *ASIP* mRNA expression from different promoters is the central mechanism regulating the so-called pigment-type switching [19,21,28]. The regulatory elements of the *ASIP* gene are most likely contained in its large 5'-flanking region. Differing from the *KIT* locus, *ASIP* does not contain a very large 3'-flanking region. Spatially and temporally regulated synthesis of eumelanin and pheomelanin enables mammals to express a wide variety of coat color patterns that are essential for e.g. camouflage or mate recognition in many wild species.

*ASIP* variants cause a wide variety of breed defining coat color phenotypes in domestic animals. *ASIP* loss of function variants, typically in the coding sequence, are responsible for recessive black in e.g. dogs [51], horses [52], and rabbits [53]. Gain of function variants in *ASIP*, such as an ectopic overexpression lead to dominant red phenotypes [54]. There are only very few examples of non-coding regulatory variants at the *ASIP* locus that have been fully characterized at the molecular level. One important example is the mouse black-and tan allele ($a^t$), which is caused by a ~6 kb retroviral-like insertion in the region of the hair cycle-specific

promoter [25]. In black and tan at mice, hairs are no longer banded and show a uniformly yellow or uniformly black pigmentation. Amplification of the entire *ASIP* gene has previously been shown to cause the white coat color in many sheep breeds [18] and the white or tan allele ($A^{Wt}$) in Saanen goats [9].

Our study confirmed the previous results and defined the exact breakpoints of the caprine $A^{Wt}$ triplication that actually comprises not only the *ASIP* gene, but also the flanking *AHCY* and *ITCH* genes. Unexpectedly, we did not observe selection signatures at the window harboring the *ASIP* gene in the Saanen and Appenzell breeds. Both of these breeds are strictly selected for uniform white (pheomelanistic) coat color and have a very high frequency of the $A^{Wt}$ allele. We think that the lack of a significant $-ZH_p$ score at *ASIP* in these two breeds is caused by at least three factors. For the calculation of the $-ZH_p$ score, we considered only SNVs with a maximum coverage of 50x in order to suppress artifacts caused by non-specific mapping of highly repetitive sequences. As the *ASIP* locus is triplicated in Saanen and Appenzell goats their pools had >50x average coverage at *ASIP* and almost no SNVs were called in the region. Furthermore, when we inspected the whole genome sequencing data from 24 individual Appenzell goats, we found that only 22 of them were $A^{Wt}/A^{Wt}$ as expected. The remaining two animals had the genotype $A^{Wt}/A^{sm}$. As $A^{Wt}$ is the most dominant allele in the series, it apparently has still not reached absolute fixation and other *ASIP* alleles are segregating at least in the Appenzell goat breed. Finally, the three copies of the triplication had several sequence differences, which were called as variable positons with a 2:1 ratio of the alleles. This provides a biological explanation why the selection signature might be weaker than expected.

We observed significant selection signatures at the *ASIP* gene in five Swiss goat breeds. In these breeds, we identified three additional non-coding CNVs in the 5'-region of *ASIP* that are likely to cause the Swiss markings ($A^{sm}$), badgerface ($A^b$) and peacock ($A^{pc}$) alleles in goats. We have to caution that the peacock phenotype has not been reported before and may be influenced by additional genes other than *ASIP*.

The corresponding coat color phenotypes are very interesting as they have characteristic patterns of eumelanistic and pheomelanistic pigmentation. Domestic goats with these alleles therefore represent a valuable resource for dissecting the precise function of individual regulatory elements in future studies. The $A^{sm}$ and $A^b$ alleles result in almost exactly inverted distributions of eumelanin and pheomelanin. Our RNA-seq data confirm that the different pigmentation patterns are caused by different levels of *ASIP* mRNA transcription at different body locations. These data also revealed that the different caprine *ASIP* gene alleles give rise to a higher number of non-coding 5'-exons compared to other mammals [25,26]. Additional data, such as Cage-seq and full-length Iso-seq data will be required for a comprehensive annotation of all possible transcription start sites and splice isoforms in goats. Such data are expected to become available soon with the advances of the FAANG project [55].

In conclusion, we identified 2,239 selection signatures in 20 diverse goat breeds with various coat color phenotypes. These selection signatures revealed six different functionally relevant CNVs underlying breed-defining coat color phenotypes in goats. The results should help to advance our mechanistic understanding of temporal and spatial regulation of transcription.

## Materials and methods

### Ethics statement

All animal experiments were performed according the local regulations. All animals in this study were examined with the consent of their owners. Sample collection was approved by the "Cantonal Committee For Animal Experiments" (Canton of Bern; permit 75/16).

## Animals

For this study, 244 female animals of 20 phenotypical diverse goat breeds and their wild ances-
tor, the bezoar, were sampled (S1 Table). Ten of the analyzed goat breeds originate from Swit-
zerland, eight from Pakistan and two from Africa. Swiss and African breeds were sampled in
Switzerland. Pakistani breeds were sampled in Pakistan. bezoar samples were from zoo ani-
mals. For the Swiss goat breeds, we selected representative animals of the breeds and excluded
any first-degree relatives. For the other goat breeds, we did not have full pedigree information
and used convenience samples. Genomic DNA was isolated from EDTA blood samples.

## Whole genome sequencing of pools (pool-seq)

Breed pools were prepared by pooling equimolar amounts of 12 animals per breed (ANG: 10,
BEZ: 8 and STG: 10). Illumina TruSeq PCR-free genomic DNA libraries with an insert size of
350 bp were prepared. Each breed pool was sequenced on one lane of an Illumina HiSeq 3000
instrument and on average 300 million 2x150 bp paired-end reads per breed pool were col-
lected (S1 Table).

## Mapping and variant calling

Adapter sequences, reads with too many Ns and low quality bases were trimmed or ultimately
discarded, if the remaining read length was < 50 bp with fastq-mcf version 1.1.2 (settings: -l 50
-S -q 20). The cleaned reads were mapped to the goat reference genome ARS1 [56] with Bur-
rows-Wheeler Aligner (BWA-MEM) algorithm version 0.7.13 [57] using the "-M" flag to mark
shorter alignments as secondary. The resulting mapping files in SAM format were converted
into BAM format and coordinate sorted using SAMtools version 1.3 [58]. A local indel realign-
ment was performed using the Genome Analysis Toolkit version 3.7 [59] with default settings.
Duplicated reads were marked, using Picard Tools version 2.2.1 (http://broadinstitute.github.
io/picard) with default settings for patterned flow cell models. Single nucleotide variants were
called using (i) Genome Analysis Toolkit UnifiedGenotyper version 3.7 [59] with the settings:
-glm SNP, -stand_call_conf 20, -out_mode EMIT_VARIANTS_ONLY and –ploidy 16/20/24
and (ii) SAMtools mpileup [60] with the settings -q 15, -Q 20, -C 50 and -B. The variants
resulting from UnifiedGenotyper were filtered for high quality variants with GATK's Variant-
Filtration tool using the generic hard-filtering recommendations available from https://
gatkforums.broadinstitute.org/gatk/discussion/6925/understanding-and-adapting-the-
generic-hard-filtering-recommendations, while the mpileup files were streamed to the PoPoo-
lation2 version 1.201 pipeline [61]. We used the scripts mpileup2sync.jar with settings - -fastq-
type sanger and - -min-qual 20 and snp-frequency-diff.pl with the settings - -min-coverage
15, - -max-coverage 50 and - -min-count 3. Both pipelines yielded similar numbers of SNV
(S2 Table).

## Sweep analysis of pool-seq data

A screen for selective sweeps was performed using the SNV file produced for each breed pool
individually by the mpileup PoPoolation2 pipeline. At each identified SNV position in the
files, we took the numbers of major ($n_{MAJ}$) and minor ($n_{MIN}$) allele counts observed in each
breed and calculated pooled heterozygosity ($H_P$) [32] with an in-house written script. The
script applies $H_P = 2\Sigma n_{MAJ}\Sigma n_{MIN}/(\Sigma n_{MAJ}+\Sigma n_{MIN})^2$ in a sliding 50% overlapping window
approach. We evaluated the results with different window sizes (25 to 300 kb) and decided on
150 kb as the most appropriate size [33, 34]. The obtained $H_P$ values for all 34,382 overlapping
150 kb windows across the whole genome were Z transformed, performing $ZH_P = (H_P-\mu H_P/$

$\sigma H_p$). Windows with a $-ZH_p \geq 4$ were retained as selective windows and adjacent or overlapping selective windows were merged into selection signatures, individually per breed. We annotated the identified selection signatures along with NCBI's *Capra hircus* Annotation Release 102 (S3 Table). In addition, to the $H_p$ calculation, we calculated weighted population $F_{ST}$ values for each SNV in a 150 kb sliding, 50% overlapping window approach. We applied the $F_{ST}$-sliding.pl script of the Popoolation2 pipeline. The script $F_{ST}$-sliding.pl was run with the settings --min-count 2 --min-coverage 4 --max-coverage 50 --window-size 150000 --step-size 75000 --suppress-noninformative and --pool-size 10:12:12:12:8:12:12:12:12:12:12:12:12:12:12:12:10:12:12:12:12. It used the previously obtained sync file of all pools combined as input and calculated weighted population pairwise $F_{st}$ values using the standard equation as shown in Hartl and Clark [62]. This resulted in 210 pairwise comparisons, from which we selected the comparisons between the 20 domesticated goat breeds with the bezoar. The obtained $F_{ST}$ values were Z transformed, performing $ZF_{ST} = (F_{ST}-\mu F_{ST}/\sigma F_{ST})$.

## Whole genome re-sequencing of individual goats and variant calling

In addition to the pool-seq experiment, we selected 120 goats from five Swiss breeds for individual whole genome re-sequencing. The 24 animals per breed included the 12 goats represented in the breed pools (S1 Table). Illumina TruSeq PCR-free DNA libraries with an insert size of 350 bp were prepared and sequenced on an Illumina NovaSeq 6000 instrument, yielding on average 240 million 2x150 bp paired-end reads per goat (S1 Table). Clean reads were produced by running fastp, version 0.12.5 [63], an ultra-fast all-in-one FASTQ preprocessor capable of trimming polyG tails, a known issue of NovaSeq reads. The cleaned reads were mapped to the ARS1 goat reference genome [56] with Burrows-Wheeler Aligner (BWA-MEM) algorithm using the "-M" flag to mark shorter alignments as secondary. The resulting SAM files were converted into BAM files and coordinate sorted using SAMtools. Duplicated reads were marked, using Picard Tools (http://broadinstitute.github.io/picard) with default settings for patterned flow cell models. The marked BAM files were streamed to GATK's Base-Recalibrator tool, supported with known SNV provided by the VarGoats consortium (http://www.goatgenome.org/vargoats.html). Subsequently, GATK's HaplotypeCaller with the settings --emitRefConfidence GVCF and -stand_call_conf 30 was used to call genome-wide variants [59]. The variant files were merged and GATK's GenotypeGVCFs was used to call variants in the 120 goats combined. As a next step, the called variants were filtered for high quality variants with GATK's VariantFiltration tool (version 3.8) using the generic hard-filtering recommendations available from https://gatkforums.broadinstitute.org/gatk/discussion/6925/understanding-and-adapting-the-generic-hard-filtering-recommendations. SnpEff [64] and NCBI's *Capra hircus* Annotation Release 102 was used to annotate the variants.

## Sweep analysis of individual goats

To calculate $H_p$ scores of the individual goats, we selected biallelic, passed SNPs per breed using GATK's SelectVariants tool (version 3.8), applying --restrictAllelesTo BIALLELIC --selectTypeToInclude SNP --sample_expressions '(APZ/BST/PFA/STG/VAG)' --maxNO-CALLnumber 0 --excludeFiltered --excludeNonVariants. This yielded on average 14.9 million SNVs per combined Swiss goat breed, comprising each 24 individually sequenced animals (S2 Table). As a next step, the VCF files containing only biallelic, passed SNPs were transformed into table format using GATK's VariantsToTable tool. This table contained only information regarding SNP position, reference allele and genotype of the 24 animals. With an in-house written Python script, we converted the table produced with GATK's VariantsToTable to

major and minor alleles and counted the number of observations. This output was then used for $H_p$ calculation as described in sweep analysis of pool-seq data.

## Manhattan plots

The $-ZH_p$ values were plotted using the function manhattan of the qqman package [65] with R [66]. Each data point represents a 150 kb window. A red horizontal line was drawn representing the chosen significance threshold of $-ZH_p \geq 4$ (corresponding to 0.8% of all windows).

## CNV analyses

Coverage plots for regions of interest were created by calculating the coverage of each base in a defined region of interest using Samtools depth -b. Additionally coverage stats across the whole genomes, including the average coverage were calculated using goleft covstats (https://github.com/brentp/goleft). Taken both results together, we plotted the coverage using R plot type h version 3.4.1 and indicated the average coverage line. Potential CNVs were also visually evaluated by inspection of the short-read alignemnts (bam-files) in the Integrative Genome Viewer (IGV) [67].

## Skin biopsies and total RNA extraction

Skin biopsies were taken from five slaughtered animals of different goat breeds (SAN, BST, GFG, PFA and VAG). Two 6 mm punch biopsies were taken from differentially pigmented body areas of each animal (S6 Table). The biopsies were immediately put in RNA*later* (Qiagen) for at least 24 h and then frozen at –20˚C. Prior to RNA extraction, the skin biopsies were homogenized mechanically with the TissueLyser II device from Qiagen. Total RNA was extracted from the homogenized tissue using the RNeasy Fibrous Tissue Mini Kit (Qiagen) according to the manufacturer's instructions. RNA quality was assessed with a FragmentAnalyzer (Advanced Analytical) and the concentration was measured using a Qubit Fluorometer (ThermoFisher Scientific).

## Whole transcriptome sequencing (RNA-seq)

From each sample, 1 μg of high quality total RNA (RIN >9) was used for library preparation with the Illumina TruSeq Stranded mRNA kit. The 10 libraries were pooled and sequenced on an S1 flow cell with 2x50 bp paired-end sequencing using an Illumina NovaSeq 6000 instrument. On average, 31.5 million paired-end reads per sample were collected (S6 Table). All reads that passed quality control were mapped to the ARS1 goat reference genome assembly using STAR aligner (version 2.6.0c) [68]. The read abundance was calculated using HTseq (version 0.9.1) [69] and a gff3 file obtained from NCBI's *Capra hircus* Annotation Release 102. We used the EdgeR package [70] to read the HTseq count data and calculated the log fold changes using the exactTest function where the biological co-efficient of variation (BCV) was set to 0.1. Trimmed mean of M (TMM) values of *ASIP* mRNA expression were determined for each sample [71].

## Supporting information

**S1 Fig. Distributions of $H_p$, $-ZH_p$, $F_{ST}$ and $-ZF_{ST}$.**
(PDF)

**S2 Fig. $-ZH_p$ scores Manhattan plots of pooled sequencing and single sequencing.**
(PDF)

**S3 Fig. Manhattan plots of −ZHp scores and ZF$_{ST}$ scores.**
(PDF)

**S4 Fig. Details of the CNV alleles.**
(PDF)

**S1 File. FASTA sequences of nine different caprine *ASIP* transcripts.**
(TXT)

**S1 Table. Read statistics and accessions of pool-seq and individual WGS data.**
(XLSX)

**S2 Table. Pool-seq and individual WGS SNV statistics.**
(XLSX)

**S3 Table. H$_p$ selection signatures per breed pool.**
(XLSX)

**S4 Table. F$_{ST}$ selection signatures per breed pool.**
(XLSX)

**S5 Table. Details of CNV alleles.**
(XLSX)

**S6 Table. Descriptive statistics and accessions of RNA-seq datasets.**
(XLSX)

## Acknowledgments

The authors are grateful to all goat owners and breeding organizations who donated samples and shared pedigree data and phenotype information of their animals. We thank Eva Andrist, Nathalie Besuchet Schmutz, Muriel Fragnière, and Sabrina Schenk for expert technical assistance, the Next Generation Sequencing Platform of the University of Bern for performing the high-throughput sequencing experiments, and the Interfaculty Bioinformatics Unit of the University of Bern for providing high performance computing infrastructure. Furthermore, we thank Christian Gazzarin for the photos and Sarah Stangl for the graphical illustrations of the different Swiss goat breeds.

## Author Contributions

**Conceptualization:** Tosso Leeb.

**Data curation:** Vidhya Jagannathan.

**Funding acquisition:** Tosso Leeb.

**Investigation:** Jan Henkel, Rashid Saif, Vidhya Jagannathan, Corinne Schmocker, Flurina Zeindler, Cord Drögemüller, Christine Flury.

**Methodology:** Vidhya Jagannathan.

**Resources:** Rashid Saif, Erika Bangerter, Ursula Herren, Dimitris Posantzis, Zafer Bulut, Philippe Ammann, Cord Drögemüller.

**Supervision:** Vidhya Jagannathan, Cord Drögemüller, Christine Flury, Tosso Leeb.

**Visualization:** Jan Henkel.

**Writing – original draft:** Jan Henkel, Tosso Leeb.

**Writing – review & editing:** Jan Henkel, Rashid Saif, Vidhya Jagannathan, Corinne Schmocker, Flurina Zeindler, Erika Bangerter, Ursula Herren, Dimitris Posantzis, Zafer Bulut, Philippe Ammann, Cord Drögemüller, Christine Flury, Tosso Leeb.

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
