## [Decision Letter · Decision Letter 0]

26 Sep 2019

Dear Dr Leeb,

Thank you very much for submitting your Research Article entitled 'Selection signatures in goats reveal copy number variants underlying breed-defining coat color phenotypes' to PLOS Genetics.

The manuscript was fully evaluated at the editorial level and by two peer reviewers. The reviewers appreciated the attention to an important problem, but raised some important concerns that should be feasible to address with additional analyses.  Based on the reviews, we will not be able to accept this version of the manuscript, but are interested in evaluating a revised version that addresses the reviewers' concerns.

If you decide to revise the manuscript for further consideration at PLOS Genetics, please aim to resubmit within the next 60 days, unless it will take extra time to address the concerns of the reviewers, in which case we would appreciate an expected resubmission date by email to plosgenetics@plos.org.

[LINK]

We are sorry that we cannot be more positive about your manuscript at this stage. Please do not hesitate to contact us if you have any concerns or questions.

Yours sincerely,

Gregory S. Barsh

Editor-in-Chief

PLOS Genetics

Gregory Copenhaver

Editor-in-Chief

PLOS Genetics

Reviewer's Responses to Questions

**Comments to the Authors:**

Reviewer #1: The authors of this study have performed pooled whole genome sequencing of 20 goat breeds and identified thousands of putative signals of selection based on reduced heterozygosity. Here they report that some of the detected putative selective sweeps involve copy number variants at KIT and ASIP associated with coat color variants. The frequent occurrence of such CNVs is interesting. Further characterization of these CNVs would make the paper even more interesting for the readers of this journal.

Specific comments

1. It would be interesting to know to which extent the copy number varies between individuals within breeds. Since the authors report pooled sequencing data it is possible that the copy numbers here are just the average and that variation in copy number may contribute to heterogeneity in phenotypes. The authors have apparently some individual sequence data that can be directly used for this and it would be easy to design TaqMan assays to explore this.

2. It would also be interesting to confirm that these are tandem duplications located at the KIT and ASIP loci and not translocated copies in other places of the genome. This can be confirmed by characterizing the duplication borders either using available short reads or by PCR. This would also allow the identification of the exact duplication breakpoints which can be used to characterize how the initial structural change occurred. Furthermore, this would allow the authors to confirm that those cases where the authors assume that different alleles share a common origin and one allele has evolved further by accumulating additional changes on the same haplotype as suggested for the two KIT alleles (Fig. 2) and for A(pc).

3. Another analysis worth doing is to explore the sequence haplotypes present in the duplicated copies, the two or more copies may be identical in sequence represent different alleles depending on how the duplication event occurred and how much sequence homogenization has taken place between the two copies. For instance, the authors report that there was not any increased homozygosity in white Saanen goats. The explanation could be that the 150 kb triplication shows sequence heterogeneity between copies which are interpreted as heterozygous position in the SNP analysis.

The authors write on page 10 that A(pc) is probably derived from the A(b) allele, this can be tested by comparing the sequence identities across the CNV region and by determining the exact duplication breakpoints.

4. As far as I understand the coat color of the Peacock goat has not been reported before. It appears very likely that the CNV reported here is contributing to the phenotype but I assume it is impossible to tell if the phenotype described as A(pc) is solely determined by an ASIP allele. It may be a combined effect of this variant in combination with one or more other loci. I recommend that the authors state that this is a preliminary description of the Apc phenotype.

5. Adalsteinsson et al described 8 ASIP alleles in goats, 5 alleles are characterized here, one of these is new, so the question is whether there are 4 more alleles not yet characterized or were these represented among the 20 different breeds but were not associated with a sweep signal and/or CNV?

6. I disagree with the statement at the bottom of page 14 that white spotting in Pak Angora and Barbari goats are likely due to insufficient expression of KIT. The lesson from the KIT literature in mouse and other species is that KIT loss-of-functions in the heterozygous state gives a mild white spotting and homozygotes are completely white and lethal or sublethal due to pleiotropic effects on hematopoiesis. In contrast, structural changes often cause dominant white spotting (e.g. patch and rump-white in mouse) most likely due to ectopic expression of KIT interfering with KIT normal function disrupting melanocyte migration/maturation. I think it is much more likely that the CNVs reported here alters the KIT expression pattern than leading to just insufficient expression.

Minor comments

1. Page 4, middle. To the best of my knowledge the ligand for MC1R is named melanocyte stimulating hormone (MSH). The 1 in melanocortin refers to that it is 1 out of 5 melanocortin receptors.

2. Page 9. There is redundancy between the figure legend and the text.

Reviewer #2: The authors used pooled sequencing and windowed selection scans to identify putative selective sweeps between 20 goat populations.

This is a very good, comprehensive analysis using an efficient method for selective sweep detection. However, there are some aspects of the approach and presentation that should be addressed.

The authors refer to a drop in pooled heterozygosity as a selective signature. This in itself is not, nor is strong FST. When combining the two, preferably coupled to a test for neutrality, one may consider a selective sweep. The authors should apply more formal tests for selection.

2,239 selection signatures, with over 100 per pool is quite high. How do the authors account for demographic factors driving divergence as opposed to positive selection, particularly since you have access to individual data from five breeds? Did they attempt other formal methods such as XP-EHH, XP-CLR, or iHS?

It would be informative to include zFST with the zHp data so readers can assess if the authors’ claims hold true. I see them referenced in Figure S1 and S3, but these were not included in the submission received by this reviewer – only supplemental tables.

How much total genomic length do the 2,239 signatures represent?

Copy number changes distort selective signals since they are collapsed in the assembly, and typically contain interlocus variation. Thus, the presence of a CNV does not mean positive selection, particularly when variants are excluded based on coverage, which would distort Hp. The authors should compare these signals to true selective sweep signals driving non-copy related evolution. Do the copy number estimates hold true for individual sequences?

Detecting sweeps is typically done using varying window sizes to test the robustness of the detected signal. Did the authors examine this, and choose 150kb based on the result? How did they determine this window size was appropriate for their data?

Identifying CNV is a separate method from identifying selective sweeps. Was there an effort to identify CNV outside sweep regions? This should be included since the authors are trying tie CNV signals to those of selective sweeps.

In sum, the discovery of these specific CNV, and their effect on ASIP are convincing and will constitute a significant contribution to the field. However, the conflation of selection and copy number is confusing, and should be clarified further. Without a comprehensive description of CNV and comparison with Hp, FST, and a more formal test for selection, it is difficult to conclude this is selection, rather than simply demography.

**Have all data underlying the figures and results presented in the manuscript been provided?**

Reviewer #1: None

Reviewer #2: No: I was not provided supplemental figures in the PDF

PLOS authors have the option to publish the peer review history of their article (what does this mean?). If published, this will include your full peer review and any attached files.

Reviewer #1: No

Reviewer #2: No

---

## [Decision Letter · Decision Letter 1]

23 Nov 2019

Dear Dr Leeb,

We are pleased to inform you that your manuscript entitled "Selection signatures in goats reveal copy number variants underlying breed-defining coat color phenotypes" has been editorially accepted for publication in PLOS Genetics. Congratulations!

Please note that Reviewer #1 has some final small comments (see below) that you may want to consider as you prepare the final version of your manuscript for the production team (the editorial team will not need to reevaluate).

Yours sincerely,

Gregory P. Copenhaver

Editor-in-Chief

PLOS Genetics

Comments from the reviewers (if applicable):

Reviewer's Responses to Questions

**Comments to the Authors:**

Reviewer #1: The authors have responded adequately to my previous comments and I have only some minor comments that they may consider.

1. Abstract. The authors may change the text to “we identified 2,239 putative selection signatures” to acknowledge the comment from reviewer 2 that a fraction of these are likely to be false positives caused by for instance genetic drift during breed formation. (The same caution may be added to the first sentence of the Discussion). They can also change to “we found strong selection signatures in a region harboring KIT” indicating that this signal stands out among the 2,239 putative signals.

2. Page 4. The detection of loci under selection in Atlantic herring was not based on loss of heterozygosity but based on allele frequency differences between populations. However, some of the loci under selection showed clear sweep signals with loss of heterozygosity.

3. Page 6, bottom line. Correct typo for TYRP1

Reviewer #2: Though there are other analyses I contend SHOULD be done, I acknowledge that the findings are significant and warrants publication.

**Have all data underlying the figures and results presented in the manuscript been provided?**

Reviewer #1: Yes

Reviewer #2: Yes

PLOS authors have the option to publish the peer review history of their article (what does this mean?). If published, this will include your full peer review and any attached files.

Reviewer #1: No

Reviewer #2: No

**Data Deposition**

http://datadryad.org/submit?journalID=pgenetics&manu=PGENETICS-D-19-01351R1

**Press Queries**

---

## [Editor Report · Acceptance letter]

10 Dec 2019

PGENETICS-D-19-01351R1 

Selection signatures in goats reveal copy number variants underlying breed-defining coat color phenotypes 

Dear Dr Leeb, 

We are pleased to inform you that your manuscript entitled "Selection signatures in goats reveal copy number variants underlying breed-defining coat color phenotypes" has been formally accepted for publication in PLOS Genetics! Your manuscript is now with our production department and you will be notified of the publication date in due course.

With kind regards,

Nicholas White

PLOS Genetics

On behalf of:
